# Going Beyond Approximation: Encoding Constraints for Explainable Multi-hop Inference via Differentiable Combinatorial Solvers

## Abstract

Integer Linear Programming (ILP) provides a viable mechanism to encode explicit and controllable assumptions about explainable multi-hop inference with natural language. However, an ILP formulation is *non-differentiable* and cannot be integrated into broader deep learning architectures. Recently, Thayaparan et al. (2022) proposed a novel methodology to integrate ILP with Transformers to achieve end-to-end differentiability for complex multi-hop inference. While this hybrid framework demonstrates to deliver better answer and explanation selection than transformer-based and ILP solvers, the neuro-symbolic integration still relies on a convex relaxation of the ILP formulation, which can produce suboptimal solutions. To improve these limitations, we propose *Diff*-Comb Explainer, a novel neuro-symbolic architecture based on *Differentiable BlackBox Combinatorial solvers* (DBCS) (Pogančić et al., 2019). Unlike existing differentiable solvers, the presented model does not require the transformation and relaxation of the explicit semantic constraints, allowing for a direct and a more efficient integration of ILP formulations. *Diff*-Comb Explainer demonstrates improved accuracy in answer and explanation selection over non-differentiable solvers, Transformers and constraint-based differentiable multi-hop inference frameworks.

## 1 Introduction

Given a question expressed in natural language, ILP-based Multi-hop Question Answering (QA) aims to construct an explanation graph of interconnected facts (i.e., natural language sentences) to support the answer (see Figure 1). This framework provides a viable mechanism to encode explicit and controllable assumptions about the structure of the inference (Khashabi et al., 2018; Khot et al., 2017; Khashabi et al., 2016). For this reason, inference based on constrained optimization is generally regarded as interpretable and transparent, providing structured explanations in support of the underlying reasoning process (Thayaparan et al., 2020).

However, ILP solvers are *non-differentiable* and cannot be integrated as part of a broader deep learning architecture (Paulus et al., 2021; Pogančić et al., 2019). Moreover, these approaches are often limited by the exclusive adoption of hard-coded heuristics for the inference and cannot be optimised end-to-end on annotated corpora to achieve performance comparable to deep learning counterparts (Thayaparan et al., 2022; Khashabi et al., 2018).

In an attempt to combine the best of both worlds, Thayaparan et al. (2022) proposed a novel neuro-symbolic framework (*Diff*-Explainer) that integrates explicit constraints with neural representations via Differentiable Convex Optimization Layers (Agrawal et al., 2019). *Diff*-Explainer combines constraint optimization solvers with Transformers-based representations, enabling end-to-end training for explainable multi-hop inference. The *non-differenitability* of ILP solvers is alleviated by approximating the constraints using semi-definite programming (Helmberg, 2000). This approximation usually requires non-trivial transformations of ILP formulations into convex optimization problems.

Since constraint-based multi-hop inference is typically framed as optimal subgraph selection via binary optimization $(0, 1)$, The semi-definite relaxation employed in *Diff*-Explainer necessitates a continuous relaxation of the discrete variables (from $\{0, 1\}$ to $[0, 1]$). While this process can provide

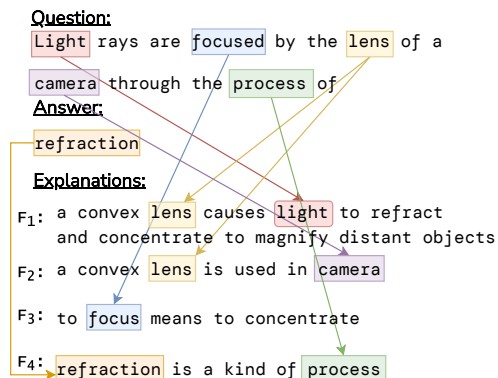

Figure 1: Example of question, answer and explanations as graph (Xie et al., 2020; Jansen et al., 2018)

tight approximations for ILP problems, this relaxation can still lead to sub-optimal solutions in practice (Yoshida, 2011; Thapper & Živný, 2018) leading to errorneous answer and explanation prediction.

To improve on these limitations, we propose *Diff*-Comb Explainer, a novel neuro-symbolic architecture based on *Differentiable BlackBox Combinatorial solvers* (DBCS) (Pogančić et al., 2019). The proposed algorithm transforms a combinatorial optimization solver into a composable building block of a neural network. DBCS achieves this by leveraging the minimisation structure of the combinatorial optimization problem, computing a gradient of continuous interpolation to address the *non-differenitability* of ILP solvers. In contrast to *Diff*-Explainer (Thayaparan et al., 2022), DBCS makes it possible to compute exact solutions for the original ILP problem under consideration, approximating the gradient.

Our experiments on multi-hop question answering with constraints adopted from ExplanationLP (Thayaparan et al., 2021) yielded an improvement of 11% over non-differentiable solvers and 2.08% over *Diff*-Explainer. Moreover, we demonstrate that the proposed approach produces more accurate and faithful explanation-based inference, outperforming non-differentiable ILP-based solvers, *Diff*-Explainer and Transformer-based approaches.

## 2 RELATED WORK

**Constraint-based multi-hop inference** ILP has been applied for structured representation (Khashabi et al., 2016) and over semi-structured representation extracted from text (Khot et al., 2017; Khashabi et al., 2018). Early approaches were unsupervised. However, recently Thayaparan et al. (2021) proposed the ExplanationLP model optimised towards answer selection via Bayesian optimisation. ExplanationLP was limited to fine-tuning only nine parameters and used pre-trained neural embedding. *Diff*-Explainer (Thayaparan et al., 2022) was the first approach to integrate constraints into a deep-learning network via Differentiable Convex Optimisation Layer (Agrawal et al., 2019) by approximating ILP constraints using Semi-definite programming. (Lovász & Schrijver, 1991).

**Hybrid reasoning with Transformers** Clark et al. (2021) proposed "soft theorem provers" operating over explicit theories in language. This hybrid reasoning solver integrates natural language rules with transformers to perform deductive reasoning. Saha et al. (2020) improved on top of it, enabling the answering of binary questions along with the proofs supporting the prediction. The multiProver (Saha et al., 2021) evolves on top of these conceptions to produce an approach that is capable of producing multiple proofs supporting the answer. While these hybrid reasoning approaches produce explainable and controllable inference, they assume the existence of natural language rules and have only been applied to synthetic datasets. On the other hand, our approach does not require extensive rules set and can tackle complex scientific and commonsense QA.

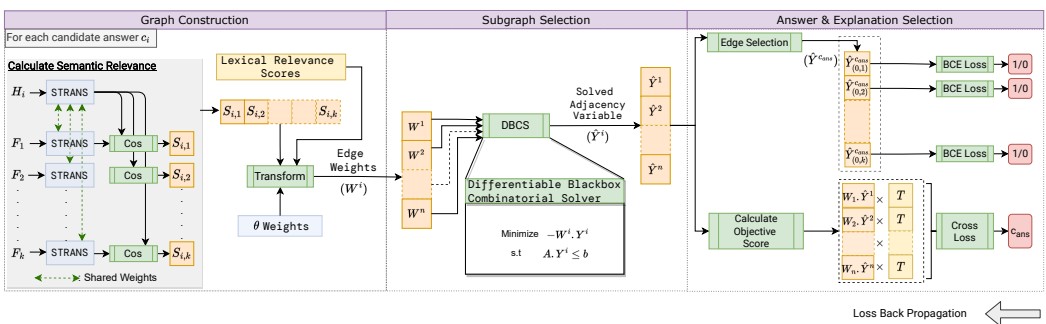

Figure 2: End-to-end architectural diagram of *Diff-Comb Explainer*. The integration of Differentiable Blackbox Combinatorial solvers will result in better explanation and answer prediction.

**Differentiable Blackbox Combinatorial Optimisation Solver** Given the following bounded integer problem:

$$\min_{x \in X} c \cdot x \qquad \text{subject to} \qquad Ax \leq b, \tag{1}$$

where $X \in \mathbb{Z}^n$, $c \in \mathbb{R}^n$, $x$ are the variables, $A = [a_1, \ldots, a_m] \in \mathbb{R}^{m \times n}$ is the matrix of constraint coefficients and $b \in \mathbb{R}^m$ is the bias term. The output of the solver $g(c)$ returns the $\arg\min_{x \in X}$ of the integer problem.

Differentiable Combinatorial Optimisation Solver (Pogančić et al., 2019) (DBCS) assumes that $A$, $b$ are constant and the task is to find the $\mathrm{d}L/\mathrm{d}c$ given global loss function $L$ with respect to solver output $x$ at a given point $\hat{x} = g(\hat{c})$. However, a small change in $c$ is *typically* not going to change the optimal ILP solution resulting in the true gradient being zero.

In order to solve this problem, the approach simplifies by considering the linearisation $f$ of $L$ at the point $\hat{x}$.

$$f(x) = L(\hat{x}) + \frac{\mathrm{d}L}{\mathrm{d}x}(\hat{x}) \cdot (x - \hat{x}) \tag{2}$$

to derive:

$$\frac{\mathrm{d}f(g(c))}{\mathrm{d}c} = \frac{\mathrm{d}L}{\mathrm{d}c} \tag{3}$$

By introducing the linearisation, the focus is now to differentiating the piecewise constant function $f(g(c))$. The approach constructs a continuous interpolation of $f(g(c))$ by function $f_\lambda(w)$. Here the hyper-parameter $\lambda > 0$ controls the trade-off between *informativeness of the gradient* and *faithfulness to the original function*.

## 3 *Diff*-Comb Explainer: Differentiable Blackbox Combinatorial Solver for Explainable Multi-Hop Inference

ILP-based QA is typically applied to multiple-choice question answering (Khashabi et al., 2018; Khot et al., 2017; Khashabi et al., 2016; Thayaparan et al., 2021). Given Question ($Q$) and the set of candidate answers $C = \{c_1, c_2, c_3, \ldots, c_n\}$ the aim is to select the correct answer $c_{ans}$.

In order to achieve this, ILP-based approaches convert question answer pairs into a list of hypothesis $H = \{h_1, h_2, h_3, \ldots, h_n\}$ (where $h_i$ is the concatenation of $Q$ with $c_i$) and typically adopt a retrieval model (e.g: BM25, FAISS (Johnson et al., 2017)), to select a list of candidate explanatory facts $F = \{f_1, f_2, f_3, \ldots, f_k\}$. Then construct a weighted graph $G = (V, E, W)$ with edge weights $W : E \rightarrow \mathbb{R}$ where $V = \{\{h_i\} \cup F\}$, edge weight $W_{ik}$ of each edge $E_{ik}$ denote how relevant a fact $f_k$ is with respect to the hypothesis $h_i$.

Given this premise, ILP-based Multi-hop QA can be defined as follows (Thayaparan et al., 2022):

**Definition 3.1** (*ILP-Based Multi-Hop QA*)**.** Find a subset $V^* \subseteq V$, $h \in V^*$ and $E^* \subseteq E$ such that the induced subgraph $G^* = (V^*, E^*)$ is connected, weight $W[G^* = (V^*, E^*)] := \sum_{e \in E^*} W(e)$ is maximal and adheres to set of constraints $M_c$ designed to emulate multi-hop inference. The

hypothesis $h_i$ with the highest subgraph weight $W[G^* = (V^*, E^*)]$ is selected to be the correct answer $c_{ans}$.

As illustrated in Figure 2, *Diff*-Comb Explainer has 3 major parts: *Graph Construction*, *Subgraph Selection* and *Answer/Explanation Selection*. In *Graph Construction*, for each candidate answer $c_i$ we construct graph $G^i = (V^i, E^i, W^i)$ where the $V^i = \{h_i\} \cup \{F\}$ and weights $W_{ik}^i$ of each edge $E_{ik}^i$ denote how relevant a fact $f_k$ is with respect to the hypothesis $h_i$. These edge weights $(W_{ik}^i)$ are calculated using a weighted $(\theta)$ sum of scores calculated using transformer-based $(STrans)$ embeddings and lexical overlap.

In the *Subgraph Selection* step, for each $G^i$ Differentiable Blackbox Combinatorial Solver (DBCS) with constraints are applied to extract subgraph $G^*$. In this paper, we adopt the constraints proposed for ExplanationLP (Thayaparan et al., 2021). ExplanationLP explicit abstraction by grouping facts into abstract and grounding facts. Abstract facts are core scientific facts that a question is attempting to test, and grounding facts link concepts in the abstract facts to specific terms in the question/answer. For example, in Figure 1 the core scientific fact is about the nature of convex lens and how they refract light $(F_1)$. Facts $F_2, F_3, F_4$ help to connect the abstract fact to the question/answer.

Finally, in *Answer/Explanation Selection* the model is to predict the correct answer $c_{ans}$ and relevant explanations $F_{exp}$. During training time, the loss is calculated based on gold answer/explanations to fine-tune the transformers $(STrans)$ and weights $(\theta)$. The rest of the section explains each of the components in detail.

## 3.1 GRAPH CONSTRUCTION

In order to facilitate grounding abstract chains, the retrieved facts $F$ are classified into *grounding* facts $F_G = \{f_1^g, f_2^g, f_3^g, ..., f_l^g\}$ and abstract facts $F_A = \{f_1^a, f_2^a, f_3^a, ..., f_m^a\}$ such that $F = F_A \cup F_G$ and $l + m = k$.

Similarly to *Diff*-Explainer (Thayaparan et al., 2022), we use two relevance scores: semantic and lexical scores, to calculate the edge weights. We use a Sentence-Transformer (STrans) (Reimers et al., 2019) bi-encoder architecture to calculate the semantic relevance. The semantic relevance score from STrans is complemented with the lexical relevance score. The semantic and lexical relevance scores are calculated as follows:

**Semantic Relevance** ($s$): Given a hypothesis $h_i$ and fact $f_j$ we compute sentence vectors of $\vec{h_i} = STrans(h_i)$ and $\vec{f_j} = STrans(f_j)$ and calculate the semantic relevance score using cosine-similarity as follows:

$$s_{ij} = S(\vec{h_i}, \vec{f_j}) = \frac{\vec{h_i} \cdot \vec{f_j}}{\|\vec{h_i}\|\|\vec{f_j}\|} \tag{4}$$

**Lexical Relevance** ($l$): The lexical relevance score of hypothesis $h_i$ and $f_j$ is given by the percentage of overlaps between unique terms (here, the function $trm$ extracts the lemmatized set of unique terms from the given text):

$$l_{ij} = L(h_i, f_j) = \frac{|trm(h_i) \cap trm(f_j)|}{max(|trm(h_i)|, |trm(f_j)|)} \tag{5}$$

Given the above scoring function, we construct the edge weights matrix ($W$) as follows:

$$W_{jk}^i = \begin{cases} -\theta_{gg}l_{jk} & (j,k) \in F_G \\ -\theta_{aa}l_{jk} & (j,k) \in F_A \\ \theta_{ga}l_{jk} & j \in F_G, k \in F_A \\ \theta_{qgl}l_{jk} + \theta_{qgs}s_{jk} & j \in F_G, k = h_i \\ \theta_{qal}l_{jk} + \theta_{qal}s_{jk} & j \in F_A, k = h_i \end{cases} \tag{6}$$

Here relevance scores are weighted by $\theta$ parameters which are clamped to $[0, 1]$.

## 3.2 SUBGRAPH SELECTION VIA DIFFERENTIABLE BLACKBOX COMBINATORIAL SOLVERS

Given the above premises, the objective function is defined as:

$$\min \quad -1(W^i \cdot Y^i) \tag{7}$$

We adopt the edge variable $Y^i \in \{0, 1\}^{(n+1) \times (n+1)}$ where $Y^i_{j,k}$ ($j \neq k$) takes the value of 1 iff edge $E^i_{jk}$ belongs to the subgraph and $Y^i_{jj}$ takes the value of 1 iff $V^i_j$ belongs to the subgraph.

Given the above variable, the constraints are defined as follows:

**Answer selection constraint** The candidate hypothesis should be part of the induced subgraph:

$$\sum_{j \in \{h_i\}} Y^i_{jj} = 1 \tag{8}$$

**Edge and Node selection constraint** If node $V^i_j$ and $V^i_k$ are selected then edges $E^i_{jk}$ and $E^i_{kj}$ will be selected. If node $V^i_j$ is selected, then edge $E_{jj}$ will also be selected:

$$Y^i_{jk} \leq Y^i_{jj} \qquad \forall\, (j, k) \in E \tag{9}$$

$$Y^i_{jk} \leq Y_{kk} \qquad \forall\, (j, k) \in E \tag{10}$$

$$Y^i_{jk} \geq Y_{jj} + Y_{kk} - 1 \qquad \forall\, (j, k) \in E \tag{11}$$

**Abstract fact selection constraint** Limit the number of abstract facts selected to $M$:

$$\sum_i Y^i_{jj} \leq M \qquad \forall j \in F_A \tag{12}$$

## 3.3 ANSWER AND EXPLANATION SELECTION

The solved adjacency variable $\hat{Y}^i$ represents the selected edges for each candidate answer choice $c_i$. Not all datasets provide gold explanations. Moreover, even when the gold explanations are available, they are only available for the correct answer with no explanations for the *wrong* answer.

In order to tackle these shortcomings and ensure end-to-end differentiability, we use the softmax ($\sigma$) of the objective score ($W^i \cdot \hat{Y}^i$) as the probability score for each choice.

We multiply each objective score $W^i \cdot \hat{Y}^i$ value by the temperature hyperparameter ($T$) to obtain soft probability distributions $\gamma^i$ (where $\gamma^i = (W^i \cdot \hat{Y}^i) \cdot T$). The aim is for the correct answer $c_{ans}$ to have the highest probability.

In order to achieve this aim, we use the cross entropy loss $l_c$ as follows to calculate the answer selection loss $\mathcal{L}_{ans}$ as follows:

$$\mathcal{L}_{ans} = l_c(\sigma(\gamma^1, \, \gamma^2, \, \cdots \gamma^n), \, c_{ans}) \tag{13}$$

If gold explanations are available, we complement $\mathcal{L}_{ans}$ with explanation loss $\mathcal{L}_{exp}$. We employ binary cross entropy loss $l_b$ between the selected explanatory facts and gold explanatory facts $F_{exp}$ for the explanatory loss as follows:

$$\mathcal{L}_{exp} = l_b(\hat{Y}^{ans}[f_1, \, f_2, \, \ldots, \, f_k], \, F_{exp}) \tag{14}$$

We calculate the total loss ($\mathcal{L}$) as weighted by hyperparameters $\lambda_{ans}, \lambda_{exp}$ as follows:

$$\mathcal{L} = \lambda_{ans}\mathcal{L}_{ans} + \lambda_{exp}\mathcal{L}_{exp} \tag{15}$$

## 4 EMPIRICAL EVALUATION

### 4.1 ANSWER AND EXPLANATION SELECTION

We use the WorldTree corpus (Xie et al., 2020) for training the evaluation of explanation and answer selection. The 4,400 question and explanations in the WorldTree corpus are split into three different subsets: *train-set*, *dev-set* and *test-set*. We use the *dev-set* to assess the explainability performance

since the explanations for *test-set* are not publicly available. The background knowledge is consists of 5000 abstract facts from the WorldTree table store (WTree) (Xie et al., 2020) and over 100,000 *is-a* grounding facts from ConceptNet (Speer et al., 2017).

**Baselines:** We use the following baselines to compare against our approach for the WorldTree corpus:

1. **BERT$_{Base}$ and BERT$_{Large}$** (Devlin et al., 2019): To use BERT for this task, we concatenate every hypothesis with $k$ retrieved facts, using the separator token [SEP]. We use the Hugging-Face (Wolf et al., 2019) implementation of *BertForSequenceClassification*, taking the prediction with the highest probability for the positive class as the correct answer.
2. **ExplanationLP**: Non-differentiable version of ExplanationLP. Using the constraints stated in Section 3, we fine-tune the $\theta$ parameters using Bayesian optimization and frozen STrans representations. This baseline aims to evaluate the impact of end-to-end fine-tuning over the non-differentiable solver.
3. *Diff*-**Explainer**: *Diff*-Explainer has already exhibited better performance over other explainable multi-hop inference approaches, including ILP-based approaches including TableILP (Khashabi et al., 2016), TupleILP (Khot et al., 2017) and graph-based neural approach PathNet (Kundu et al., 2019). Similar to our approach, we use ExplanationLP constraints with *Diff*-Explainer. We use similar hyperparameters and knowledge base used in Thayaparan et al. (2022).

**Metrics** The answer selection is evaluated using accuracy. For evaluation of explanation selection, we use Precision@$K$. In addition to Precision@$K$, we introduce two new metrics to evaluate the truthfulness of the answer selection to the underlying inference. The metrics are as follows:

**Explanatory Consistency@$K$**: Question/answer pair with similar explanations indicates similar underlying inference (Atanasova et al., 2022). The expectation is that similar underlying inference would produce similar explanations (Valentino et al., 2021; 2022). Given a test question $Q_t$ and retrieved explanations $E_t$ we find set of Questions $Q_t^s = \{Q_t^1, Q_t^2, \ldots\}$ with at least $K$ overlap gold explanations along with the retrieved explanations $E_t^s = \{e_t^1, e_t^2, \ldots\}$. Given this premise, Explanatory Consistency@$K$ is defined as follows:

$$\frac{\sum_{e_t^i \in E_t^s} [e_t^i \in E_t]}{\sum_{e_t^i \in E_t^s} |e_t^i|} \tag{16}$$

Explanatory Consistency measures out of questions/answer pairs with at least $K$ similar gold explanations and how many of them share a common retrieved explanation.

**Faithfulness**: The aim is to measure how much percentage of the correct prediction is derived from correct inference and wrong prediction is derived from wrong inference over the entire set. Let's say that the set of questions correctly answered as $A_{Q_c}$, wrongly answered questions $A_{Q_w}$, set of questions with at least one correctly retrieved explanation as $A_{Q_1}$ and set of questions where no correctly retrieved explanations $A_{Q_0}$. Given this premise, Faithfulness is defined as follows:

$$\frac{|A_{Q_w} \cap A_{Q_0}| + |A_{Q_c} \cap A_{Q_1}|}{|A_{Q_c} \cup A_{Q_w}|} \tag{17}$$

A higher faithfulness implies that the underlying inference process is reflected in the final answer prediction.

Table 1 illustrates the explanation and answer selection performance of *Diff*-Comb Explainer and the baselines. We report scores for *Diff*-Comb Explainer trained for only the answer and optimised jointly for answer and explanation selection.

Since BERT does not provide explanations, we use facts retrieved from the fact retrieval for the best $k$ configuration ($k = 3$) as explanations. We also report the scores for BERT without explanations.

We draw the following conclusions from the results obtained in Table 1 (The performance increase here are expressed in absolute terms):

**(1)** *Diff*-Comb Explainer improves answer selection performance over the non-differentiable solver by 9.47% with optimising only on answer selection and 10.89% with optimising on answer and explanation selection. This observation underlines the impact of the end-to-end fine-tuning framework. We can also observe that strong supervision with optimising explanation selection yields better performance than weak supervision with answer selection.

| Model | Explanation Selection (*dev*) | | | | | | Answer Selection (*test*) |
|---|---|---|---|---|---|---|---|
| | Precision | | Explanatory Consistency | | | Faithfulness | |
| | @2 | @1 | @3 | @2 | @1 | | |
| **Baselines** | | | | | | | |
| BERT$_{Base}$ | - | - | - | - | - | - | 45.43 |
| BERT$_{Large}$ | - | - | - | - | - | - | 49.63 |
| Fact Retrieval (FR) Only | 30.19 | 38.49 | 21.42 | 15.69 | 11.64 | - | - |
| BERT$_{Base}$ + FR | - | - | - | - | - | 52.65 | 58.06 |
| BERT$_{Large}$ + FR | - | - | - | - | - | 51.23 | 59.32 |
| ExplanationLP | 40.41 | 51.99 | 29.04 | 14.14 | 11.79 | 71.11 | 62.57 |
| *Diff*-Explainer | 41.91 | 56.77 | 39.04 | 20.64 | 17.01 | 72.22 | 71.48 |
| ***Diff*-Comb Explainer** | | | | | | | |
| - Answer selection only | 45.75 | 61.01 | **49.04** | 29.99 | 18.88 | 73.37 | 72.04 |
| - Answer and explanation selection | **47.57** | **63.23** | 43.33 | **33.36** | **20.71** | **74.47** | **73.46** |

Table 1: Comparison of explanation and answer selection of *Diff*-Comb Explainer against other baselines. Explanation Selection was carried out on the *dev* set as the *test* explanation was not public available.

**(2)** *Diff*-Comb Explainer outperforms the best transformer-based model by 14.14% for answer selection. This increase in performance demonstrates that integrating constraints with transformer-based architectures leads to better performance.

**(3)** *Diff*-Comb Explainer outperform the best *Diff*-Explainer configuration (answer and explanation selection) by 0.56% even in the weak supervision setting (answer only optimization). We also outperform *Diff*-Explainer by 1.98% in the best setting.

**(4)** *Diff*-Comb Explainer is better for selecting relevant explanations over the other constraint-based solvers. *Diff*-Comb Explainer outperforms the non-differentiable solver at Precision@K by 8.41% ($k = 1$) and 6.05% ($k = 2$). We also outperform *Diff*-Explainer by 3.63% ($k = 1$) and 4.55% ($k = 2$). The improvement of Precision@$K$ over the Fact Retrieval only (demonstrated with BERT + FR) by 16.98% ($k = 1$) and 24.74% ($k = 2$) underlines the robustness of our approach to noise propagated by the upstream fact retrieval.

**(5)** Our models also exhibit higher Explanatory Consistency over the other solvers. This performance shows that the optimization model is learning and applying consistent inference across different instances. We also outperform the fact retrieval model which was also a transformer-based model trained on gold explanations.

**(6)** Answer prediction by *black-box* models like BERT do not reflect the explanation provided. This fact is indicated by the low Faithfulness score obtained by both BERT$_{Base}$/BERT$_{Large}$. In contrast, the high constraint-based solver's Faithfulness scores emphasise how the underlying inference reflects on the final prediction. In particular, our approach performs better than the non-differentiable models and *Diff*-Explainer.

In summary, despite the fact that *Diff*-Explainer and *Diff*-Comb Explainer approaches use the same set of constraints, our model yields better performance, indicating that accurate predictions generated by ILP solvers are better than approximated sub-optimal results.

## 4.2 QUALITATIVE ANALYSIS

Table 2 illustrates some of the explanations extracted for ExplanationLP, *Diff*-Explainer and *Diff*-Comb Explainer. Both explanations and answer predictions in Question (1) are entirely correct for our model. In this example, both ExplanationLP and *Diff*-Explainer have failed to retrieve any correct explanations or predict the correct answer. Both the approaches are distracted by the strong lexical overlaps with the wrong answer.

| |
|---|
| **Question (1):** Which measurement is best expressed in light-years?: **Correct Answer:** the distance between stars in the Milky Way. |
| ExplanationLP |
| **Answer**: the time it takes for planets to complete their orbits. **Explanations**: *(i)* a complete revolution; orbit of a planet around its star takes 1; one planetary year, *(ii)* a light-year is used for describing long distances |
| *Diff*-Explainer |
| **Answer**: the time it takes for planets to complete their orbits. **Explanations**: *(i)* a light-year is used for describing long distances, *(ii)* light year is a measure of the distance light travels in one year |
| *Diff*-Comb Explainer |
| **Answer**: the distance between stars in the Milky Way. **Explanations**: *(i)* light years are a astronomy unit used for measuring length, *(ii)* stars are located light years apart from each other |
| **Question (2):** Which type of precipitation consists of frozen rain drops?: **Correct Answer:** sleet. |
| ExplanationLP |
| **Answer**: snow. **Explanations**: *(i)* precipitation is when snow fall from clouds to the Earth, *(ii)* snow falls |
| *Diff*-Explainer |
| **Answer**: sleet. **Explanations**: *(i)* snow falls, *(ii)* precipitation is when water falls from the sky |
| *Diff*-Comb Explainer |
| **Answer**: sleet. **Explanations**: *(i)* sleet is when raindrops freeze as they fall, *(ii)* sleet is made of ice |
| **Question (3):** Most of the mass of the atom consists of?: **Correct Answer:** protons and neutrons. |
| ExplanationLP |
| **Answer**: neutrons and electrons. **Explanations**: *(i)* neutrons have more mass than an electron, *(ii)* neutrons have more mass than an electron |
| *Diff*-Explainer |
| **Answer**: protons and neutrons. **Explanations**: *(i)* the atomic mass is made of the number of protons and neutrons, *(ii)* precipitation is when water falls from the sky |
| *Diff*-Comb Explainer |
| **Answer**: protons and neutrons. **Explanations**: *(i)* the atomic mass is made of the number of protons and neutrons, *(ii)* precipitation is when water falls from the sky |

Table 2: Example of predicted answers and explanations (Only *CENTRAL* explanations) obtained from our model with different levels of fine-tuning.

Question (2) at least one explanation is correct and a correct answer prediction for our model. In the example provided, *Diff*-Explainer provides the correct answer prediction with both the retrieved facts not being explanatory. *Diff*-Explainer arrives at the correct answer prediction with no explanation addressing the correct answer.

In Question (3) both our model and *Diff*-Explainer provide the correct answer but with both facts not being explanations. The aforementioned qualitative (Question 1 and 2) and quantitative measures (Explanatory Consistency@$K$, Faithfulness) indicate how the underlying explanatory inference results in the correct prediction; there are cases where false inference still leads to the correct answer with our model as well. In this case, the inference is distracted by the strong lexical overlaps irrelevant to the question.

However, from the qualitative analysis, we can conclude that the explainable inference that happens with our model is more robust and coherent when compared to the *Diff*-Explainer and non-differentiable models.

### 4.3 KNOWLEDGE AGGREGATION WITH INCREASING DISTRACTORS

One of the key characteristics identified by Thayaparan et al. (2022) is the robustness of *Diff*-Explainer to distracting noise. In order to evaluate if our model also exhibits the same characteristics, we ran our model for the increasing number of retrieved facts $k$ and plotted the answer selection accuracy for WorldTree in Figure 3.

As illustrated in the Figure, similar to *Diff*-Explainer, our approach performance remains stable with increasing distractors. We also continue to outperform *Diff*-Explainer across all sets of $k$.

BERT performance drops drastically with increasing distractors. This phenomenon is in line with existing work (Thayaparan et al., 2022; Yadav et al., 2019a). We hypothesise that with increasing distractors, BERT overfits quickly with spurious inference correlation. On the other hand, our ap-

| Model | Explainable | Accuracy |
|---|---|---|
| BERT$_{Large}$ | No | 35.11 |
| IR Solver (Clark et al., 2016) | Yes | 20.26 |
| TupleILP (Khot et al., 2017) | Yes | 23.83 |
| TableILP (Khashabi et al., 2016) | Yes | 26.97 |
| ExplanationLP (Thayaparan et al., 2021) | Yes | 40.21 |
| DGEM (Clark et al., 2016) | Partial | 27.11 |
| KG$^2$ (Zhang et al., 2018) | Partial | 31.70 |
| ET-RR (Ni et al., 2019) | Partial | 36.61 |
| Unsupervised AHE (Yadav et al., 2019b) | Partial | 33.87 |
| Supervised AHE (Yadav et al., 2019b) | Partial | 34.47 |
| AutoRocc (Yadav et al., 2019a) | Partial | 41.24 |
| *Diff*-Explainer (ExplanationLP) (Thayaparan et al., 2022) | Yes | **42.95** |
| *Diff*-Comb Explainer (ExplanationLP) | Yes | **43.21** |

Table 3: ARC challenge scores compared with other Fully or Partially explainable approaches trained *only* on the ARC dataset.

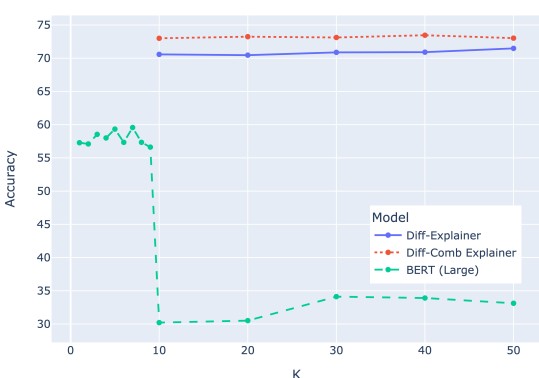

Figure 3: Accuracy for different number of retrieved facts.

proach circumvents this problem with the inductive bias provided by the constraint optimization layer.

### 4.4 COMPARING ANSWER SELECTION WITH ARC BASELINES

Table 3 presents a comparison of publicly reported baselines on the ARC Challenge-Corpus (Clark et al., 2018) and our approach. These questions have proven to be challenging to answer for other LP-based question answering and neural approaches. While models such as UnifiedQA (Khashabi et al., 2020) and AristoBERT (Xu et al., 2021) have demonstrated performance of 81.14 and 68.95, they have been trained on other question-answering datasets, including RACE (Lai et al., 2017). Moreover, despite its performance, UnifiedQA does not provide explanations supporting its inference.

In Table 3, to provide a rigorous comparison, we only list models that have been trained *only* on the ARC corpus and provides explanations supporting its inference to ensure fair comparison. Here the explainability column indicates if the model delivers an explanation for the predicted answer. A subset of the models produces evidence for the answer but remains intrinsically black-box. These models have been marked as *Partial*. As illustrated in the Table 3, *Diff*-Comb Explainer outperforms the best non-differentiable constraint-solver model (ExplanationLP) by 2.8%. We also outperform a transformer-only model AutoRocc by 1.97%. While our improvement over *Diff*-Explainer is small, we still demonstrate performance improvements for answer selection. On top of performances obtained for explanation and answer selection with WorldTree corpus, we have also established better performances than leaderboard approaches.

## 5 CONCLUSION

This paper proposed a novel framework for encoding explicit and controllable assumptions as part of an end-to-end learning framework for explainable multi-hop inference using Differentiable Black-box Combinatorial Solvers (Pogančić et al., 2019). We empirically demonstrated improved answer and explanation selection performance compared with the existing differentiable constraint-based solver for multi-hop inference (Thayaparan et al., 2022). We also demonstrated performance gain and increased robustness to noise when combining constraints with transformer-based architectures. In this paper, we adopted the constraints of ExplanationLP, but it is possible to encode more complex inference constraints within the model.

*Diff*-Comb Explainer builds on previous work by Thayaparan et al. (2022) and investigates the combination of symbolic knowledge (expressed via constraints) with neural representations. We hope this work will encourage researchers to encode different domain-specific priors, leading to more robust, transparent and controllable neuro-symbolic inference models for NLP.

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

## 6 APPENDIX

The pseudo-code to train *Diff*-Comb Explainer end-to-end is summarized in Algorithm 1.

**Experimental Setup:** We employ the following experimental setup:

- *Sentence Transformer Model*: ALL-MPNET-BASE-V2 (Song et al., 2020).
- *Fact retrieval representation*: ALL-MPNET-BASE-V2 trained with gold explanations of WorldTree Corpus to achieve a Mean Average Precision of 40.11 in the dev-set.
- *Fact retrieval*: FAISS retrieval (Johnson et al., 2017) using pre-cached representations.
- *Background knowledge*: 5000 abstract facts from the WorldTree table store (WTree) and over 100,000 *is-a* grounding facts from ConceptNet (CNet) (Speer et al., 2017).
- The experiments were carried out for $k = \{1, 2, 3, 5, 10, 20, 30, 40, 50\}$ and the best configuration for each model is selected.
- The hyperparameters $\lambda, \lambda_{ans}, \lambda_{exp}, T$ were fine-tuned for 50 epochs using the Adpative Experimentation Platform.
- $M$=2 for ExplanationLP, *Diff*-Explainer and *Diff*-Comb Explainer.

### 6.1 EXTERNAL CODE-BASES

- Differentiable Blackbox Combinatorial Solvers Examples: https://github.com/martius-lab/blackbox-differentiation-combinatorial-solvers
- Sentence Transformer code-base: https://huggingface.co/sentence-transformers/all-mpnet-base-v2

---

**Algorithm 1:** Training *Diff*-Comb Explainer

---

**Data:** $A$, $b \leftarrow$ Multi-hop Inference Constraints
**Data:** $f_w \leftarrow$ Graph weight Function
**Data:** $\lambda \leftarrow$ Hyperparameter for DBCS interpolation
$epoch \leftarrow 0$;
**while** *epoch $\leq$ max_epochs* **do**
    **foreach** $h_i \in H$ **do**
        $G^i \leftarrow$ fact-graph-construction($h_i$, $F$);
        $l^i \leftarrow L(h_i, F)$;
        $\theta \leftarrow clamp([0, 1])$;
        $\vec{F} \leftarrow STrans(F)$;
        $\vec{h_i} \leftarrow STrans(h_i)$;
        $s^i \leftarrow S(\vec{h_i}, \vec{F})$;
        $W^i \leftarrow f_w(s^i, l^i; \theta)$;
        $\hat{Y}^i \leftarrow DBCS(-W^i, A, b; \lambda)$;
        $\gamma^i \leftarrow (W \cdot \hat{Y}^i) \cdot T$;
    **end**
    $\mathcal{L}_{ans} = l_c(\sigma(\gamma^1, \gamma^2, \cdots \gamma^n), c_{ans})$;
    **if** *$F_{exp}$ is available* **then**
        $\mathcal{L}_{exp} = l_b(\hat{Y}^{ans}[f_1, f_2, \ldots, f_k], F_{exp})$;
        $\mathcal{L} = \lambda_{ans}\mathcal{L}_{ans} + \lambda_{exp}\mathcal{L}_{exp}$;
    **else**
        $\mathcal{L} = \mathcal{L}_{ans}$;
    **end**
    update $\theta$, $STrans$ using AdamW optimizer by minimizing $loss$;
    $epoch \leftarrow epoch + 1$;
**end**
**Result:** Store best $\theta$ and $STrans$

---

## 6.2 INTEGER LINEAR PROGRAMMING OPTIMIZATION

The components of the linear programming system is as follows:

- Solver: Gurobi Optimization `https://www.gurobi.com/products/gurobi-optimizer/`

The hyperparatemers used in the ILP constraints:

- Maximum number of abstract facts ($M$): 2

Infrastructures used:

- CPU Cores: 32
- CPU Model: Intel(R) Core(TM) i7-6700 CPU @ 3.40GHz
- Memory: 128GB

## 6.3 HYPERPARAMETERS

For *Diff*-Comb Explainer we had to fine-tune hyperparameters $\lambda, \lambda_{ans}, \lambda_{exp}, T$. We fine-tune for 50 epochs using the Adpative Experimentation Platform with seed of 42.

The bounds of the hyperparameters are as follows:

- $\lambda$: $[100, 300]$
- $\lambda_{exp}$: $[0.0, 1.0]$
- $\lambda_{ans}$: $[0.0, 1.0]$
- $T$: $[1e-2, 100]$

The hyperparameters adopted for our approach are as follows:

- $\lambda$: 152

- $\lambda_{exp}$: 0.72
- $\lambda_{ans}$: 0.99
- $T$: 8.77
- max epochs: 8
- gradient accumulation steps: 1
- learning rate: 1e-5
- weight decay: 0.0
- adam epsilon: 1e-8
- warmup steps: 0
- max grad norm: 1.0
- seed: 42

The hyperparameters adopted for BERT are as follows:

- gradient accumulation steps: 1
- learning rate: 1e-5
- weight decay: 0.0
- adam epsilon: 1e-8
- warmup steps: 0
- max grad norm: 1.0
- seed: 42

We fine-tuned using 4 Tesla V100 GPUs for 10 epochs in total with batch size 32 for $Base$ and 16 for $Large$.

## 6.4   DATA

**WorldTree Dataset**: Data can be obtained from: `http://cognitiveai.org/explanationbank/`

**ARC-Challenge Dataset**: `https://allenai.org/data/arc`. Only used the Challenge split.

