# OpenReview forum: "Going Beyond Approximation: Encoding  Constraints for Explainable Multi-hop Inference via Differentiable Combinatorial Solvers"
_ICLR.cc/2023/Conference — Submitted to ICLR 2023_

### Official Review · Reviewer_eQxh · 2022-10-24

**Confidence:** 3
**Correctness:** 3
**Technical Novelty And Significance:** 2
**Empirical Novelty And Significance:** 2
**Recommendation:** 5

**Clarity, Quality, Novelty And Reproducibility:**

The writing could be improved a bit (I did notice typos and in some cases, the meaning of sentences was hard to read). The novel contribution could be highlighted better an particularly analyzed more as to why it impacts explainability since that was not so clear to me. Novelty in a bit limited though since it is based on a recent model that has a similar approach though uses a different type of ILP solver to integrate into the tranformer-model. Since standard benchmarks are used, they should be reproducible, however several hyperparameters are mentioned in the ILP formulation. The effect of varying these are not clear.

**Strength And Weaknesses:**

Strengths
- Empirical results show improvements in explanation accuracy over existing state-of-the-art DiffExplainer
- Using DBCS seems to be a new way to integrate symbolic constraints into transformers and the same could be applicable in other models as well.

Weakness
- Novelty seems somewhat limited since the ideas seem to be a similar to the ideas in Diffexplainer (Except the use of DBCS). I am not sure if there is only one dataset that is typically used for evaluation (since ARC does not measure explanations) to show more significant impact of the proposed method
- In general, solving the ILP exactly may be hard (for different types of constraints), so is this approach generalizable or does it work due to the specific type of constraints formulate for the multiple-choice answering problem


**Summary Of The Paper:**

The paper describes an approach to add ILP constraints into transformer-based models. The main idea is to allow for end-to-end training using a differentiable solver (DBCS) to solve the ILP. Here, DBCS produces can produce exact solutions to the ILP to approximate the gradient. The main hypothesis is that using these exact solutions to the ILP generates better explanations in multi-hop inference.

Specifically, the approach is described in the context multiple-choice question answering. Explanations are extracted in the form of weighted graphs. The weights encode the importance of a fact for a hypothesis. The subgraph that represents the explanations are selected using DBCS.
The transformer-based model (STrans) is used for computing embeddings needed to compute weights for the graph in the formulated ILP problem.

Experiments are performed using the world tree corpus and results are reported for explanations on the dev-dataset (since test-set explanations are not available). Comparisons are made with BERT, a non-differentiable version of the approach (thus, no end-to-end training) and a recent similar approach (DiffExplainer) that uses approximate ILP solutions using transformations to a convex optimization problem. New metrics to measure consistency and faithfulness of explanations are also evaluated on all compared models.

**Summary Of The Review:**

The paper proposes an interesting idea to improve explainability in Multiple-choice question answering using ILPs that give us exact solutions and can be integrated into transformers for end-to-end training. Overall, the novelty and generalizability of the model seem to be the main weaknesses.

---

### Official Review · Reviewer_NWPP · 2022-10-24

**Confidence:** 4
**Correctness:** 4
**Technical Novelty And Significance:** 2
**Empirical Novelty And Significance:** 2
**Recommendation:** 5

**Clarity, Quality, Novelty And Reproducibility:**

### Clarity
- The paper has good clarity, it is well-written and easy to follow. The architecture overview is very helpful.
### Quality
- Overall the paper has relatively high quality, all the experiments appear to be reasonably chosen, but some questions are still open (see weaknesses).
### Novelty
- The novelty of the paper is limited. It extremely closely follows the previous work Diff-Explainer, and basically only replaces the semi-definite relaxation with the true ILP, using the DBCS method to differentiate it.
### Reproducibility
- The authors provide the pseudo-code for training the architecture and the hyperparameters that were used to produce the reported results. Code that produces the reported results is not provided.

**Strength And Weaknesses:**

### Strengths:
- The paper is well written, easy to understand, with a good presentation.
- The demonstrated success of a hybrid approach for combining symbolic knowledge in the form of ILP constraints with neural representations opens exciting possibilities for further applications in natural language processing.

### Weaknesses:
- Technical novelty: The paper is extremely similar to Diff-Explainer [1], only the semi-definite programming relaxation of the ILP is replaced with the unrelexad ILP, and DBCS [2] is used to differentiate it. This limits the theoretical contribution to plugging one existing method into another, which is very incremental.
- Empirical novelty: Currently the paper reports results only on experiments from [1], and additionally adds the novel consistency and faithfullness metrics. This is a reasonable empirical contribution, however, it is in my opinion not enough to outweight the limited theoretical novelty.
- Differentiation of the combinatorial solver: Equation 13 only depends on the optimal value of the optimization problem (which is continuous), not the optimal solution (which is discrete). Therefore it is continuously differentiable w.r.t. the parameters $W$, and complicated methods like DBCS are not required here (in contrast to differentiation of Equation 14). Was this taken into account in differentiating Equation 13 or are the reported results produced by using the informative gradient replacement from DBCS? In the latter case, it would be important to report as a baseline the results of differentiating Equation 13 without the use of DBCS.

### Additional questions:
- Is there an interpretation for the performance drop from using supervision on the explanation selection in the consistency@3 column of Table 1?
- What are the runtimes for the experiments? Solving a relxation of an ILP can be much faster than solving the true ILP, is the slight performance increase worth the additional runtime?


[1]: M. Thayaparan, M. Valentino, D. Ferreira, J. Rozanova, and A. Freitas. Diff-Explainer: Differentiable Convex Optimization for Explainable Multi-hop Inference. In Transactions of the Association for Computational Linguistics, 2022.

[2]: M. Pogancic, A. Paulus, V. Musil, G. Martius, and M. Rolinek. Differentiation of blackbox combinatorial solvers. In International Conference on Learning Representations, 2019.

**Summary Of The Paper:**

This paper concerns the problem of question answering via multi-hop inference, where multiple separate facts need to be taken into account to answer a complex question. To allow for interpretable structured explanations of the reasoning process, this and previous work explicitly encode multi-hop inference in the constraints of an Integer Linear Program.

Given a question, a potential answer, and a set of supporting facts, all the components are encoded as the nodes of an explanation graph, on which the edge weights are computed based on lexical overlap and transformer-predicted semantic relevance. The graph is then used as the input to a discrete optimization procedure that explicitly encodes multi-hop inference in its constraints, yielding as the optimal solution a subset of the provided facts to support the potential answer. Repeating this procedure for each potential answer allows to select the most promising answer along with its supporting facts.

The main difficulty of training such an architecture end-to-end lies in the non-informative jacobian of discrete optimization algorithms. To alleviate this, previous work relies on differentiable semi-definite programming relaxations. However, these can lead to sub-optimal solutions. In contrast, the authors of this work approach the differentiability issue by relying on Differentiable BlackBox Combinatorial Solvers, a technique for computing an informative replacement for the gradient. The resulting proposed framework is called Diff-Comb Explainer.

It is empirically tested on answer and explanation selection tasks with two datasets. The proposed method is shown to outperform previous methods in terms of all relevant metrics, and is also shown to be robust to an increased number of distracting facts. Some failure cases of the presented method and its competitors are also highlighted in a qualitative analysis.

**Summary Of The Review:**

While exhibiting high clarity and reasonable quality of the proposed framework, along with a good evaluation, the presented novelty is limited. I suggest to add an additional contribution, potentially further theoretical or empirical insights into the discrepancy arising from using an ILP vs. a semi-definite relaxation.

---

### Official Review · Reviewer_nF5J · 2022-10-25

**Confidence:** 4
**Correctness:** 3
**Technical Novelty And Significance:** 2
**Empirical Novelty And Significance:** 2
**Recommendation:** 3

**Clarity, Quality, Novelty And Reproducibility:**

**Clarity:**

The paper is not very well written. Some sentences are not well-formed (see below for an example).  There are some gaps/inconsistencies in the notation.

1.   Below eqn 1, $X \in Z^n$, instead it should be $X \subseteq Z^n$.
2.   Below eqn 3, what is $w$ in $f_\lambda(w)$ ?
3.   Eqn 7, need to define $W^i_{jj}$ for $j = h_i$. Below eqn 7, it should be (k+1) x (k+1) instead of (n+1) x (n+1) while defining $Y^i$
4.   Eqn 14, is incorrect. BCE computation requires probabilities for the explanatory facts. From where do we get those? $\hat{Y}^{ans}$ contains only binary assignments of the edges. Guessing from fig 2, BCE  should be computed using the edge weights $W^{ans}[0,j]$ for the $j^{th}$ fact.
5. Eqn 16.. Is $e^i_t$ a fact or a set of facts? The numerator suggests that it should be a fact, as belongingness to retrieved explanations ($E_t$, which is a set of facts) is checked, but the denominator suggests that it should be a set of facts as its cardinality is being computed.
6. “ExplanationLP explicit abstraction by grouping facts into abstract and grounding facts.”  (pg. 4): not sure what it means.


**Novelty:**

Given that Diff Explainer is already a published work, the novelty in Diff Comb Explainer is somewhat limited.

**Reproducibility:**

Given that the gains over Diff Explainer are within 2-3 points, I would encourage the authors to repeat experiments using multiple seeds and confirm whether the gain is more than the standard deviation.



**Strength And Weaknesses:**

This paper demonstrates that using differentiable black-box solvers gives a slight improvement over using SDP relaxation for making ILPs differentiable.

The idea of using an ILP to solve the exact same subgraph selection problem for precisely the same task has already been explored in Diff Explainer.  The only difference is in the choice of technique to make the ILP differentiable: Diff Explainer uses an SDP relaxation to make ILP differentiable, whereas Diff Comb Explainer uses differentiable blackbox solvers, which is the same as using CombOptNet (Paulus et al 2021) with known constraints and learnable cost.  CombOptNet has been discussed in the Diff Explainer paper, and this work is just replacing  SDP relaxation with CombOptNet with known constraints (similar to blackbox differentiation).

[Paulus et al 2021] CombOptNet: Fit the Right NP-Hard Problem by Learning Integer Programming Constraints


**Summary Of The Paper:**

This paper proposes “Diff Comb Explainer”: a neuro-symbolic architecture for selecting an answer (from a given set of candidates) as well as explanations (from a corpus of facts) for a multiple-choice QA task.
Each candidate answer is first converted into a hypothesis by concatenating the question to it, and then scoring a hypothesis is cast as a subgraph selection problem that is solved via a standard ILP for subgraph selection problem with an added constraint that hypothesis node should be a part of the subgraph. The graph contains a node for the hypothesis and k more nodes, one each for a retrieved fact from the corpus. The edges represent lexical and semantic similarities between the nodes. The facts corresponding to the nodes selected in the subgraph represent the explanations, and the objective function of the ILP is the total edge weight of the subgraph.



**Summary Of The Review:**

The idea of creating a neuro-symbolic architecture by making an ILP differentiable for the task of multiple choice QA (with explanations) has already been demonstrated by Diff Explainer. This work is simply replacing the SDP relaxation with Differentiable Black Box Solver and showing slight improvement. Notably, CombOptNet, which essentially uses a black box solver for ILPs and makes it differentiable, has already been mentioned/discussed in the Diff Explainer paper, but not chosen over SDP relaxation technique to make ILP differentiable. In light of all this, I believe that the novelty of this work is somewhat limited.

---

### Official Review · Reviewer_sAEZ · 2022-10-28

**Confidence:** 2
**Correctness:** 2
**Technical Novelty And Significance:** 2
**Empirical Novelty And Significance:** 2
**Recommendation:** 5

**Clarity, Quality, Novelty And Reproducibility:**

### Novelty and Quality

As a reader unfamiliar with the ILP-based Multi-hop QA, I'm having difficulties assessing the novelty and the quality of this work. The proposed method is based on a very specific model (Diff-Explainer) which seems to use a unique stack of techniques (integer programming, multi-hop inference and etc.) in solving the QA task. Judging from the empirical results, the Diff-Comb indeed outperforms the prior work, but since the setting and prior work are not properly presented in the paper (see clarity), it is difficult for me to tell which part of the proposed method is intellectually novel.


### Clarity


I find the contents very difficult to follow. In particular,
- I'm unable to relate the defined $H,F,G$ to the example in Fig 1. It would be nice if the authors could give a running example to illustrate the concepts.
- Def 3.1 is taken from the prior work without elaboration. For example, it is unclear to me what is the purpose of finding the induced subgraph and how this is done in prior work. More importantly, I cannot relate the terms "multi-hop inference" and "ILP" to the contents. In fact, these terms can be misleading in the QA context as there are many QA methods that utilize multi-hop reasoning on knowledge graphs and one of the techniques used here is referred to as inductive logic programming (ILP).
- Given that this work builds on top of a specific approach to the QA task with a specific technical stack, it would be nice to formally introduce the big picture and the intuitions behind them before presenting the proposed method.

Other confusions:
- What are abstract facts mentioned in Section 3?
- What does it mean "to facilitate grounding abstract chains"? What is "grounding" mentioned in 3.1? Is it related to the grounding notion in first-order logic?


**Strength And Weaknesses:**

Strength
- The approach of generating explanations for QA tasks via graph construction is interesting

Weaknesses
- The scope of this work is to improve a specific type of approach to QA tasks and the impact could be limited
- The paper is difficult to follow for readers unfamiliar with this line of research


**Summary Of The Paper:**

This work proposes a differentiable model, namely Diff-Comb Explainer, that solves the integer linear programming based multi-hop QA problem. The proposed model utilizes the differentiable blackbox combinatorial solver, which solves the ILP problem in a differentiable fashion. In the experiments, the proposed method outperforms the baselines in terms of explanation and answer selection.

**Summary Of The Review:**

This work focuses on improving the performance of QA tasks with explanations based on an existing approach. The proposed method relies on a unique stack of techniques and demonstrates better performance than the prior work in the experiments. That said, the impact of this work is limited. There might be some novelties for this particular line of research but I'm not familiar with the literature to be certain.

---

> ### Comment · Reviewer_sAEZ · 2022-12-04
> **Updates**
>
> After reading others' reviews, it confirms my initial impression that the proposed work is closely related to the prior work Diff Explainer, and the contributions seem minor. Given that the authors haven't responded to my concerns, I am lowering my score.

---

### Decision · Program_Chairs · 2023-01-20

**Decision:**

Reject

**Justification For Why Not Higher Score:**

too close to prior work

**Justification For Why Not Lower Score:**

N/A

**Metareview: Summary, Strengths And Weaknesses:**

++ Better performance than prior work
-- No reviewer willing to champion the paper
-- Novelty is unclear (and significance compared to Diff Explainer)